# Echocardiographic Left Ventricular Function in the Third Year After COVID-19 Hospitalization: A Follow-Up Pilot Study in South-East of Romania

**DOI:** 10.3390/medicina61020333

**Published:** 2025-02-14

**Authors:** Constantin-Marinel Vlase, Cristian Gutu, Roxana Elena Bogdan Goroftei, Andreea Boghean, Traian Florin Daniel Iordachi, Anca-Adriana Arbune, Manuela Arbune

**Affiliations:** 1Medical Clinic Department, Dunarea de Jos University, 800008 Galati, Romania; constantin.vlase@ugal.ro (C.-M.V.); manuela.arbune@ugal.ro (M.A.); 2“Dr. Aristide Serfioti” Military Emergency Hospital, 800008 Galati, Romania; daniel.iordachi@gmail.com; 3Clinic Emergency Children Hospital, 800487 Galati, Romania; 4Doctoral School of Biomedical Sciences, Dunarea de Jos University, 800008 Galati, Romania; 5Multidisciplinary Integrated Center for Dermatological Interface Research, 800010 Galati, Romania; anca.arbune@icfundeni.ro; 6Neurology Department, Fundeni Clinical Institute, 077086 Bucharest, Romania; 7Infectious Diseases Clinic I, Infectious Diseases Clinic Hospital Galati, 800179 Galati, Romania

**Keywords:** COVID-19, echocardiography, left ventricular disfunction, global longitudinal strain impairment

## Abstract

*Background and Objectives*: Cardiac involvement in COVID-19 has been confirmed during the acute stage of the infection. However, the prevalence and spectrum of post-infectious cardiac dysfunction remain incompletely clarified. The objective of our study was to evaluate the frequency of echocardiographic changes 2 years after hospitalization for moderate and severe COVID-19 in patients with no previously known cardiac pathology. *Material and Methods*: We conducted a retrospective cohort study analyzing severity markers of COVID-19 infection and echocardiographic parameters assessed ≥2 years after the acute illness, based on recent guideline recommended algorithm for echocardiographic diagnostic of left ventricular (LV) dysfunction. *Results*: The study included 50 Caucasian patients, 60% male, 54% aged < 65 years, and 32% with severe forms of the disease. The primary comorbidities were hypertension, obesity, and diabetes. COVID-19 severity correlated with the computed tomography (CT) lung lesion score and a neutrophil-to-lymphocyte ratio >6 but was not associated with post-COVID-19 echocardiographic changes. Left ventricular ejection fraction (LVEF) was reduced in only 18% of cases, but global longitudinal strain (GLS) impairment was observed in 46% of patients, contributing to the LV systolic subclinical dysfunction in 61%. Impaired LV diastolic disfunction with normal pressure filling was present in 30.61% of cases and with elevated pressure 10.2%. *Conclusions*: COVID-19 is an independent predictive factor for GLS impairment, which can indicate myocardial contractile dysfunction, even in patients with asymptomatic heart disease. This underscores the importance of regular echocardiographic monitoring for patients recovering from moderate to severe COVID-19.

## 1. Introduction

Cardiovascular manifestations are part of the clinical expression of infection with Severe Acute Respiratory Syndrome Coronavirus 2 (SARS-CoV-2). The persistence of cardiac lesions and sequelae can be categorized under post-COVID-19 syndrome, but they remain insufficiently documented [1]. The most common cardiac manifestation of acute COVID-19 infection is elevated cardiac troponin levels. Echocardiographic studies have revealed changes such as right ventricular dysfunction (26.3%), left ventricular dysfunction (18.4%), diastolic dysfunction (13.2%), and pericardial effusion (7.2%) [2]. Within the first 6 months post-COVID-19 infection, chest pain occurs in 5% of patients and palpitations in 9%, although the frequency of these symptoms can increase, with chest pain affecting up to 17%, palpitations up to 20%, and exertional dyspnea up to 30% of cases [3,4,5]. Post-COVID syndrome is a newly recognized clinical entity, defined as a multisystemic condition characterized by persistent symptoms following the acute infection episode, including fatigue, shortness of breath, and chest pain. These symptoms may be attributed to heart failure, myocarditis, or ventricular dysfunction resulting from COVID-19 [6,7,8]. Post-COVID-19 syndrome with persistent symptoms after recovery from SARS-CoV-2 coronavirus infection encompasses major cardiac complications such as myocardial injury, heart failure, arrhythmias, vascular lesions, and thromboses [9].

Long-COVID syndrome is a multisystemic condition that persists for more than 12 weeks after acute SARS-CoV-2 (COVID-19) infection and cannot be explained by alternative diagnoses. The mechanism of this syndrome is multifactorial, involving the immune response, levels of anti-SARS-CoV-2 neutralizing antibodies, direct effects of the virus, complications of severe forms of the disease, post-traumatic stress, and oxidative stress. Long-COVID cardiac involvement results from several prolonged effects. These include pro-inflammatory cytokines, tumor necrosis factor, interleukin 1 and interleukin 6; redox imbalance and autonomic cardiac dysfunction are also factors. Together, these can lead to extended consequences [10]. Other contributing factors include chronic systemic inflammation. This is often associated with obesity or diabetes mellitus. Undiagnosed pre-existing heart conditions can also play a role. Genetic factors are another possibility, although they remain insufficiently clarified [5]. The incidence of arrhythmias in Long-COVID is unknown, although many individuals report palpitations. Myocarditis and pericarditis can have chronic courses. Coronary artery aneurysms, aortic aneurysms, atherosclerosis, and venous or arterial thrombotic disease are also potential complications. These changes may predispose to arrhythmias and life-threatening acute coronary syndromes [7,8].

To monitor heart inflammation, one of the most effective and sensitive tests is cardiac magnetic resonance imaging (MRI), which shows inflammation in up to 60% of patients more than 2 months after a COVID-19 diagnosis [11,12]. Speckle-tracking echocardiography (STE) can also be used as a prognostic indicator in acute COVID-19 infection [13].

We hypothesized that the ventricular dysfunctions identified post-COVID-19 may persist and reveal specific echocardiographic features. The study objective is to determine the frequency of echocardiographic changes observed 2 years after hospitalization for moderate and severe COVID-19. These results could contribute to the characterization of post-COVID syndrome in the South-Eastern region of Romania, as no previous reports on this subject are available.

## 2. Materials and Methods

We conducted a retrospective cohort study on the impact of moderate and severe forms of COVID-19 on cardiac function, assessed through transthoracic echocardiography (TTE). The study was developed at “Dr. Aristide Serfioti” Military Emergency Hospital in Galati, situated in the South-East of Romania. From November 2020 to May 2021 the hospital treated COVID-19 patients. The selection criteria for the study were age over 20 years, hospitalization with moderate and severe COVID-19 confirmed by an RT-PCR test between November 2020 and May 2021, and pulmonary involvement of at least 25% on chest CT imaging. The study adhered to WHO criteria for classifying the severity of COVID-19. Moderate cases were defined as the presence of clinical signs of pneumonia (fever, cough, dyspnea, shortness of breath) but with SpO_2_ ≥ 90% on room air. Severe cases were defined by at least one of the following: respiratory rate >30 breaths/min, severe respiratory distress, or SpO_2_ < 90% on room air [14,15,16].

From the hospital database, we identified 270 cases with “diagnostic related-group code” U07.1 [17]. We excluded the cases requiring orotracheal intubation or other specific intensive care interventions (critical forms of COVID-19), those transferred to other hospitals, and those who died in the hospital. From 204 randomized patients recovered after moderate and severe COVID-19, we have selected 102 cases, of whom 96 had available contact information. A total of 79 patients/contacts responded to the phone call, but we found that 5 of them had died since COVID-19 hospitalization. Based on the medical history, we excluded four cases with pre-existing cardiac diseases diagnosed before COVID-19 episode. Fifty-three patients agreed to participate in the planned echocardiography visit, based on a scheduled invitation by phone call, between July 2023 and October 2023. Three examinations were invalidated due to poor acoustic windows, resulting in a study sample of 50 cases (Figure 1).

We collected demographic data (age, gender, urban or rural residence, and education level) as well as declared smoking status (defined as smoking within 28 days prior to the COVID-19 diagnosis). We recorded comorbidities known at the time of hospitalization for COVID-19, including hypertension, obesity (defined as a body mass index >35 kg/m^2^), diabetes mellitus, and chronic renal, hepatic, cerebrovascular diseases.

From the hospitalization records of COVID-19, we evaluated the degree of pulmonary involvement as described in the CT imaging report (%), based on the COVID-19 imaging interpretation protocol. CT imaging was performed using a “Somatom Scope Siemens—Power 32 slice device employing Iterative Reconstruction Image Space (IRIS)” techniques. The images were assessed to identify typical COVID-19 pneumonia features, such as ground-glass opacities, unilateral or bilateral, subpleural localization, predominantly in the lower lobes, with peripheral or posterior distribution, and progression to consolidation. The pulmonary lobes (three in the right lung and two in the left lung) were visually examined and the percentage of involvement in each lobe was recorded using the following scoring system: Score 1 (<5%); Score 2 (5–25%); Score 3 (25–50%); Score 4 (50–75%); Score 5 (>75%). The total score was the sum of the partial scores, ranging from 0 to 25. In our study, we used the global CT severity score [18,19,20,21].

Laboratory data collected at the time of hospitalization for the COVID-19 episode included leukocyte count, C-reactive protein, erythrocyte sedimentation rate, D-dimer levels, serum ferritin, and serum creatinine. We calculated the peripheral blood neutrophil-to-lymphocyte ratio (NLR) as the absolute neutrophil count divided by the absolute lymphocyte count, using it as a marker of systemic inflammation and a predictor for patients with COVID-19 infection. The normal NLR values for 95% of healthy adults range between 0.78 and 3.53 [22,23,24].

Echocardiography was preceded by an electrocardiogram. Cardiac rhythms were evaluated by analyzing electrocardiographic recordings taken at the follow-up visit. We categorized the results as sinus rhythm or atrial fibrillation. Sinus rhythm was identified by the presence of a P wave, QRS complex, and T wave with normal morphological and duration characteristics [25]. Atrial fibrillation was diagnosed according to the definition in the guidelines of the European Society of Cardiology [26].

Transthoracic echocardiography (TTE) was performed by an expert cardiologist specializing in echocardiographic examination and interpretation, following the methodology outlined in echocardiography guidelines. The procedure utilized the “FUJIFILM LISENDO 880 Cardiovascular Ultrasound System” (System version 00-5.0.0; Application 2DTT), including advanced analysis applications [27]. The following echocardiographic parameters were recorded: E/A ratio, E/e’ ratio, indexed left atrial volume (LAVi), S’ wave, e’ velocity, left ventricular ejection fraction (LVEF), and global longitudinal strain (GLS).

Left ventricle (LV) diastolic function was evaluated using mitral E wave, A wave, and e’ medium velocities [27,28]. The trans mitral Doppler E wave represents early LV filling from rapid relaxation. Trans mitral A wave in sinus rhythm corresponds to the atrial contraction in late diastole and LV filling. The E/A is the ratio of early and late LV diastolic filling. Typically, E/A > 1 values are found in healthy young people, but natural aging results in a shift in E/A to < 1. The Early Diastolic Mitral Annular Velocity, denoted as e’, was measured at the mitral annulus (septal and lateral walls) and was used for evaluating diastolic function and estimating left ventricular filling pressures. It is compared with E wave to calculate the E/e’ ratio, an indicator of diastolic dysfunction [27]. The left atrial volume was also used as a marker of diastolic dysfunction. It was estimated by the biplane Simpson’s Method of Disks and was indexed to the body surface area (LAVi), with normal range lower than 34 mL/m^2^ [27,29].

The systolic function was estimated by LV ejection fraction (LVEF), systolic waves (S), and global longitudinal strain (GLS) to define myocardial performance (Table A1, Table A2 and Table A3). The values of LVEF, defined as the fraction of chamber volume ejected during systole relative to the blood volume in the ventricle at the end of diastole, was interpreted according to the criteria established by the American Society of Echocardiography and the European Association of Cardiovascular Imaging [30]. The cut-off value for “normal” LVEF >55% was adjusted by gender. Systolic waves (S) were measured as part of tissue Doppler imaging at the same locations as e’, estimating specifically the systolic longitudinal contraction of the myocardium. The normal range of the S wave was ≥9 cm/s measured on a lateral mitral annulus and ≥7 cm/s measured on a septal mitral annulus. Reduced S was an indicator of impaired systolic function. Global longitudinal strain (GLS) is defined as the change in heart dimensions during cardiac activity, evaluated by measuring the left ventricular length during systole. However, GLS mirrors the complex interferences of disturbances in left ventricular (LV) relaxation, restorative forces, myocyte elongation loading, and atrial function, culminating in increased LV filling pressures [31,32]. GLS is expressed as a percentage (%) of the relative change in the length of the left ventricular myocardium between diastole and end-systole, compared to the myocardium length at the end of diastole. Normally, peak GLS is expected to be in the range of −18%, but variable values are influenced by age and gender [33]. An LVEF of  <50% or absolute GLS < 16% [27] is consistent with impaired LVDF. LVEF ≥ 50% or absolute GLS ≥ 16% do not confirm normal diastolic function and wider consideration of secondary parameters may be required [27,29].

Considering the reference values for age groups and gender, we classified the results of the echocardiographic examination into the following categories: normal, low/increased. The left ventricular diastolic function was evaluated considering the British Society of Echocardiography algorithm recommendations [27]. Systolic function was considered impaired when LVEF was lower than 50% and GLSAs was classified according to updated LV diastolic function criteria. The recommendations call for classification as either normal, indeterminate, or abnormal based on the presence or absence of abnormalities in average E/e’ > 14, septal e’ < 7 cm/s or lateral e’ < 10 cm/s, LA volume index > 34 mL/m^2^, and tricuspid regurgitation velocity <280 cm/s. [28]. We used e’ < 7 cm/s for statistical analysis. Tricuspid regurgitation velocity was not available in our study. Instead, we employed an adapted algorithm based on two out of three parameters of left ventricular diastolic function [27].

Statistical analysis used the XLSTAT version 2022.1. The data collected were formulated in a MS Excel table and statistically analyzed, according to the numerical or qualitative types of variables. We have applied statistical tests such as unpaired t test/Mann–Whitney test for the quantitative variables and Chi-square test/Fisher’s exact test for qualitative variables. The two-sample *t*-test was used to compare whether there is difference between two independent groups. The uncertainty associated with a statistical measure, such as the odds ratio (OR), was assessed using 95% confidence intervals (CIs). Confidence intervals provide a range of values within which the true value of the measure is likely to fall. In the context of odds ratios, the null value is 1. If the confidence interval includes 1, it indicates that there is no statistically significant association between the variables at the chosen confidence level. A value of *p* <  0.05 was considered statistically significant.

The study was approved by the Institutional Bord of Ethical Committee corresponding code No.109/M1-544/28 February 2023.

## 3. Results

### 3.1. Epidemiological and Clinical Characteristics of the COVID-19 Hospitalized Patients

In total, fifty study patients aged between 36 and 80 years, with an average age of 60.5 ± 10.37 years and a male-to-female ratio of 1.5 participated in the study. Other demographic characteristics included 96% living in urban areas and 52% having attained upper secondary education. Additionally, 22% were reported as active smokers. Regarding occupation, 50% of the patients were retired, 40% were employed, and 10% were unemployed. The most common co-morbidities were hypertension (42%), obesity (36%), and diabetes (18%) (Table 1).

### 3.2. Imaging and Biological Markers of Severity/Prognosis of the COVID-19 Episode

The classification according to the severity of COVID-19 identified 71.4% moderate forms and 28.6% severe forms of diseases. The imagistic scores considered the extent of the computer tomographic pulmonary lesions ranging between 25% and 80%, with an average of 49 ± 17.46 and median 50%. Severe disease was defined as a CT score >50%. Electrocardiographic data showed sinus rhythm in all but one case, where atrial fibrillation was observed during the post-COVID-19 evaluation. Blood pressure during the COVID-19 episode ranged from 85 to 170 mmHg systolic and 50 to 100 mmHg diastolic, remaining within similar ranges post-COVID (90–170 mmHg systolic and 50–95 mmHg diastolic).

The main biological changes observed in hospitalized patients with moderate and severe forms of COVID-19 were the hyperinflammation (CRP, ESR) and hypercoagulation (D-dimers, ferritin). The wide variation in the ranges of these parameters indicate the heterogeneity of the biological response and the systemic impact of the disease in hospitalized patients (Table 2). The neutrophil-to-lymphocyte ratio (NLR) is a marker of inflammatory stress, showing high values in most of our patients and predicting a high risk of death related to COVID-19. However, NLR is not a specific biomarker for COVID-19, as high values also predict poor prognosis in other conditions, including solid tumors, and chronic respiratory, cardiovascular, or renal diseases [24].

The severe form of COVID-19 was associated with increased CT sores > 50%, NLR > 6, and blood level of CRP > 75 ng/mL, but we found no correlation with gender, older age over 65, smoking, or chronic co-morbidities as hypertension, obesity, and diabetes (Table A4).

### 3.3. Echocardiographic Characteristics After COVID-19

The most common findings on transthoracic echocardiographic examination were an increased E/A’ ratio (54%) and decreased S wave (54%). Global cardiac function was reduced in 46% of all patients (Table 3).

The single case found with atrial fibrillation (AF) was evaluated according to the specific algorithm for the estimation of LVFP in patients with AF and we found a single available criterion (septal E/e’ > 11). However, none of the other parameters of the algorithm has sufficient accuracy to be considered adequate stand-alone measures for the assessment of LVDF [27,28,34,35].

Following the algorithm for the estimation of LVFP in patients with SR, we assessed 49 patients of which 38.8% had normal LVSF and 61.2% with impaired LVSF. After the second step of the algorithm, we found 22.4% patients had normal systolic and diastolic function, 10.2% patients with impaired LV function and elevated filling pressures, and 30.61% with impaired diastolic function with normal filling pressures. Insufficient criteria for estimating LV function were identified in over a third of patients [18], who were subsequently classified as “equivocal”. Among these, the majority (11 out of 18) had impaired left ventricular systolic function (LVSF) (Figure 2).

In summary, 40.81% of patients developed a disfunction of the LV dysfunction by the third year after COVID-19 hospitalization, while 22.4% maintained normal values of echocardiographic parameters. However, the equivocal cases require additional parameters to be evaluated to complete the diagnostic criteria and the frequency of LV dysfunctions could be higher.

### 3.4. Comparation of LV Dysfunction and Normal LV Function Related to COVID-19 Severity

Considering the high percentage of LV dysfunction, we hypothesized a role of severe COVID-19 on altering the cardiac structure and function. We compared the COVID-19 characteristics, as echocardiographic normal or impaired function, but no significant correlation was found (Table 4).

## 4. Discussion

The results of our study evidenced a high frequency of newly developed LV dysfunction in the third year after COVID-19 hospitalization, in patients with no previous cardiac history, when it could be expected that some acute COVID-related lesions would have recovered. Compared with other reports on this topic, which refer to the echocardiographic aspects in the first year post-COVID, the assessment of our study covers a longer interval after COVID-19. There was no correlation between cardiac changes and the severity of the infectious episode, characterized by the prognostic markers, as the extension of the CT lung lesions, level of inflammation, or the NLR. We could hypothesize that SARS-CoV-2 cardiac damage is not necessarily dependent on the severity of disease, but rather that subclinical viral impairment of myocardial tissues could be persistent and progressive. Nevertheless, other overlapping conditions could be involved in the development of LV dysfunction, such as aging, lifestyle or other acute viral infections.

The two-dimensional GLS of the left ventricle (LV) was the most significant echographic finding, suggesting that it could be persistent or progressive from the early infection. This statement is based on the results of a previous study conducted 3 months post-COVID, compared to a control group, demonstrating that GLS of the LV was more impaired parameter. Further multivariate analysis revealed that COVID-19 infection was the only independent determinant of reduced LV GLS [36,37].

The frequency of post-COVID-19 cardiac manifestations, as estimated by different studies, varies significantly, leading to controversial recommendations about the utility of post-infectious echocardiographic monitoring [38].

The subclinical dysfunction of both ventricles and the risk of long-term cardiovascular events was reported in a retrospective observational study, on 42 subjects with persistent symptoms after 1-month post-COVID-19 [36].

The prospective studies in 1-year post-COVID-19, including patients infected with alfa and beta SARS-CoV-2 variants, with mild (74%), moderate (4.8%), severe (18.1%), and critical (2.9%) forms of disease, found persistent levels of troponin and NT-proBNP, while LVEF had not significantly decreased [39]. A multicentric study in patients with no known cardiac diseases reported that post-COVID-19 cardiac involvement is rare, mostly in men, more frequent in men, with persistent symptoms in 41.8% of cases. The frequency of echocardiographic changes was 8.2%: 5.7% decrease in GLS, 3% low LVEF, and 1.1% motility dysfunction [40].

The serial echocardiographic evaluation in the Polish multidisciplinary CRACoV-HHS study evidenced elevation of cardiac biomarkers in the acute phase of COVID-19, but normal LVEF and no significant cardiac dysfunction after 1 year, suggesting that the acute cardiac consequences of COVID-19 are associated with systemic inflammation and hemodynamic stress in patients without preexisting conditions [41].

A cross-sectional study conducted between 2021 and 2023 in a Long-COVID clinic found a frequency of LV dysfunction of 24% and 8% associated RV dysfunction. Like our results, ventricular dysfunction was not influenced by gender, income, race, CRP, or the association of Long-COVID symptoms, with the only predictor being age [42].

Left and right ventricular global longitudinal strain are the criteria for the diagnosis of myocardial dysfunction, when the LVEF is normal. These were assessed during acute COVID-19 and post-COVID-19. Survivors after the acute episode of COVID-19 have normal deformation parameters, but are weaker than in healthy people from the control groups. Until now, there have been no available epidemiological studies on the prevalence of myocardial dysfunction. GLS reversion capacity seems better for RV than for LV, probably due to different pathological mechanisms, with partially reversible changes in the stromal bed in the case of RV dysfunction, compared to scarring and persistent microvascular dysfunction in the case of LV dysfunction. The pathogenic mechanisms are not sufficiently clarified, the decrease in deformation parameters having complex causes [13].

The analysis of myocardial deformation through global longitudinal strain (GLS) has become an increasingly utilized method in recent years for the early detection of global and regional myocardial function abnormalities and for differentiating types of cardiomyopathies [43]. This marker is more sensitive than ejection fraction for predicting cardiovascular events. Considering the contractile kinetics of the left ventricle (LV), longitudinal fibers are responsible for 60% of the stroke volume and appear more sensitive to ischemia due to their higher oxygen consumption, explaining GLS changes even in subclinical forms of various cardiac pathologies, such as drug-induced toxic cardiomyopathy in oncological, hematological, rheumatological, renal, and infectious diseases [33,44]. GLS has been used as a criterion for cardiological treatment in a randomized clinical trial involving oncology patients (the SUCCOUR study), with favorable results regarding LVEF preservation, although current evidence is insufficient to use GLS as a decisive factor for oncological cardiotoxic medication management [45]. The results of a recent meta-analysis highlight the utility of cardiac screening in apparently healthy individuals through echocardiography supplemented with strain imaging, avoiding invasive investigations. Extrapolating these data, and in agreement with the results of our study, GLS can be used as an early indicator of cardiac damage in patients with a history of severe COVID-19 [46]. The clinical applications of GLS allow for the detection of subclinical contractility disorders and the prevention of irreversible myocardial damage through cardiovascular therapies [43].

### The Limits of the Study

The first limitation of the study is the absence of prior echocardiographic evaluations before the acute COVID-19 episode and during hospitalization, which would have allowed us to exclude underdiagnosed cardiac diseases related to the patients’ medical history. Additionally, we did not have a control group comprising healthy individuals of the same age and gender from the general Romanian population without a history of COVID-19, which would have enabled comparison of the echocardiographic parameters. The sample size of the pilot study is relatively small and statistical significance of the results could be in short supply.

MRI was not available at our center to confirm the echocardiographic diagnosis. The study relied solely on echocardiographic evaluations, which may underestimate cardiac impairment compared to MRI findings [47]. Since the onset of COVID-19, echocardiographic examination protocols and normal reference values have undergone changes, complicating the interpretation of the data [48].

Several parameters were unavailable for the study and cases with incomplete evaluations could not be included in the statistical analysis. The retrospective nature of the study further limited the systematic collection of biological markers with prognostic relevance for cardiac impairment. Finally, the small patient cohort size affects the statistical robustness and consistency of the analysis.

## 5. Conclusions

The echocardiography findings in the third year post-COVID-19 in patients without prior cardiac disease revealed a significant number of patients with subclinical myocardial dysfunction, as revealed by the decrease in GLS and the presence of diastolic dysfunction. Systemic inflammation and hemodynamic changes associated with pulmonary impairment during COVID-19 may affect myocardial function, although no correlations were found between infection severity markers and LV dysfunction, as found in our small sample group. Whether these findings are due to the constraint of such a small sample or are consistent in the general population remains to be established by conducting further larger studies. GLS could serve as a screening marker for better understanding the spectrum of post-COVID-19 cardiological impairment, and, as our study found, may reveal further understanding in the development of further cardiovascular complications. The mechanisms of post-COVID-19 myocardial impairment are complex, overlap with natural dysfunction, remain insufficiently clarified, and require long-term, extensive epidemiological studies. Our small study has just set the premises for further evaluation—whether asymptomatic myocardial dysfunction is more prevalent in general population, even long term after the initial COVID disease, and whether this dysfunction can lead to the development of clinical manifestations of heart failure.

## Figures and Tables

**Figure 1 medicina-61-00333-f001:**
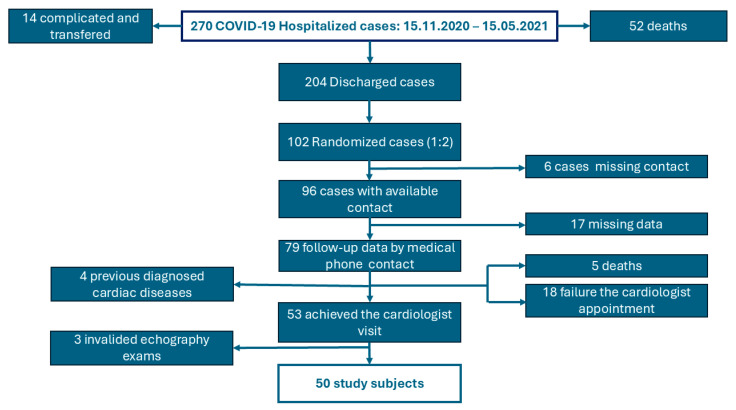
Flow chart of the study-patient selection.

**Figure 2 medicina-61-00333-f002:**
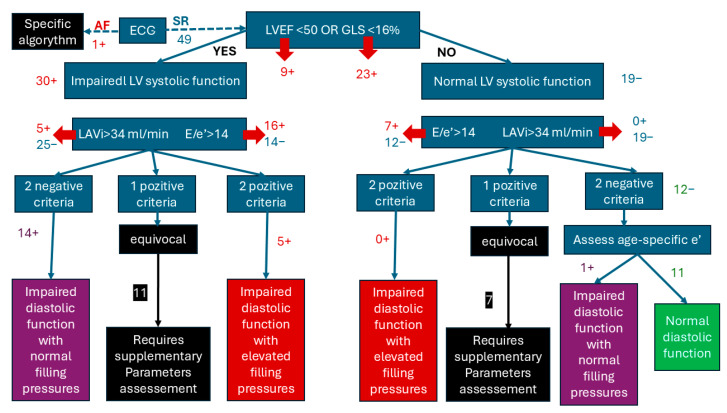
Algorithm for the estimation of LVSF after COVID-19 in previous normal cardiac history patients.

**Table 1 medicina-61-00333-t001:** Clinical and epidemiological characteristics of the patients hospitalized with COVID-19 (small sample *t*-tests).

N = 50	Option	n	%	*p*	CI 0.95
Sex	Male Female	30 20	60% 40%	0.202	0.264; 0.549
Age	>65 years old <65 years old	23 27	46% 54%	0.671	0.393; 0.682
Living area	Rural Urban	2 48	4% 96%	<0.001	0.862; 0.995
Education level	2nd Education Upper 2nd Education High Education	13 26 11	26% 52% 22%	0.887	0.374; 0.664
Professional status	Retired Employed Unemployed	25 20 5	50% 40% 10%	<0.001	0.033; 0.219
Smoking	Yes No	11 39	22% 78%	<0.001	0.64; 0.884
Hypertension	Yes No	21 29	42% 58%	0.322	0.281; 0.568
Obesity	Yes No	18 32	36% 64%	0.064	0.491; 0.771
Diabetes	Yes No	9 41	18% 82%	<0.001	0.685; 0.914
CVD	Yes No	4 46	8% 92%	<0.001	0.842; 0.997
CLD	Yes No	3 47	6% 94%	<0.001	0.872; 1.007
CKD	Yes No	1 49	2% 98%	<0.001	0.893; 0.999
COPD	Yes No	1 49	2% 98%	<0.001	0.893; 0.999
Other CID	Yes No	8 42	16% 84%	<0.001	0.017; 0.292

Legend: 2nd Education: 8 years formal education; Upper 2nd Education: 12 years formal education; High Education: bachelor’s or higher degree; CHD: cerebral-vascular disease; CLD: chronic liver disease; CKD: chronic kidney disease; COPD: chronic obstructive pulmonary disease; CD: chronic inflammatory disease (thyroid; arthritis).

**Table 2 medicina-61-00333-t002:** Biological characteristics of patients hospitalized with COVID-19.

	NV	Hospitalized Patients with COVID-19			
		No	Average ± SD	Median	Min; Max
WBC [/µL]	4–10 × 10^3^/µL	49	6.798 ± 3.25	6.2	1.5; 16.1
NLR	<3.5	48	6.51 ± 4.79	5.49	1.15; 24.97
CRP [mg/dL]	<0.1	41	75.67 ± 58.36	56.6	3.4; 186.1
ESR [mm/h]	0–10	36	77.75 ± 34.80	75.5	12; 140
D-DIMERS [ng/mL]	0–200	21	939.49 ± 687.72	785.72	97.25; 3290
FERITINE [µg/L]	<300 (♂) <150 (♀)	35	666.18 ± 377.97	696.56	58.8; 1200
CREATININE [mg/dL]	0.5–1.1 (♂) 0.6–1.2 (♀)	47	0.90 ± 0.40	0.84	0.44; 3.28

Legend: WBC: white blood cells; No: number of available values; NLR: neutrophil/lymphocyte ratio; NV: normal values; SD: standard deviation.

**Table 3 medicina-61-00333-t003:** Echocardiographic characteristics in the third year post-COVID-19 on hospitalized patients.

N = 50	Normal	Low	High	*p* *	CI
LAVi	43 (86%)	-	7 (14%)	<0.001	0.761; 0.958
E/A	23 (46%)	-	27 (54%)	0.571	0.398; 0.681
E/e’	29 (38%)	-	21 (42%)	0.257	0.279; 0.560
e’	43 (86%)	7 (14%)		<0.001	0.761; 0.958
S	23 (46%)	27 (54%)	-	0.571	0.398; 0.681
LVEF	41 (82%)	9 (18%)	-	<0.001	0.710; 0.929
GLS	27 (54%)	23 (46%)	-	0.571	0.398; 0.681

Legend: * Two sample tests for Difference between Proportions; LVEF: left ventricle ejection fraction; LAVi: left atrial volume indexed; E/A: ratio between E wave and A wave (peak velocity from LV in early diastole and diastole filling); S: left ventricular systolic velocity.

**Table 4 medicina-61-00333-t004:** Comparison of patients with normal and impaired LVF according to COVID-19 characteristics.

N = 49		Impaired LVF	Normal LVF	OR	*p*	CI 0.95
LVSF	CT-score > 50	7	7	0.521	0.307	0.149; 1.821
	N/Ly > 6	11	9	0.796	0.703	0.246; 2.575
	CRP > 75	8	9	0.404	0.137	0.122; 1.338
Impaired diastolic filling pressure	CT-score > 50	4	6	0.25	0.076	0.053; 1.157
	N/Ly > 6	8	6	0.666	0.580	0.158; 2.812
	CRP > 75	6	5	0.600	0.501	0.135; 2.657

## Data Availability

The raw data supporting the conclusions of this article will be made available by the authors on request.

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
