# Peer review of "Echocardiographic Left Ventricular Function in the Third Year After COVID-19 Hospitalization: A Follow-Up Pilot Study in South-East of Romania"

_medicina, 2025, doi:10.3390/medicina61020333_

Round 1
Reviewer 1 Report
Comments and Suggestions for Authors
Authors conclude that COVID-19 is an independent predictive factor for GLS impairment, which can indicate myocardial contractile dysfunction, even in patients with asymptomatic heart disease. However I am not sure this retrospective study over only 50 patients may add something to medical literature, also considering comorbidities over 2 years of follow-up, leading to 18% of reduction of Left ventricular ejection fraction (LVEF)
Author Response
Dear Reviewer,
Thank you very much for your time in carefully reviewing the manuscript and for your thoughtful feedback raising important points. We appreciate the opportunity to clarify and address these concerns.
Comments and Suggestions for Authors
Authors conclude that COVID-19 is an independent predictive factor for GLS impairment, which can indicate myocardial contractile dysfunction, even in patients with asymptomatic heart disease. However, I am not sure this retrospective study over only 50 patients may add something to medical literature, also considering comorbidities over 2 years of follow-up, leading to 18% of reduction of Left ventricular ejection fraction (LVEF).
Respond:
- Sample Size and Study Design: While our pilot- study includes a relatively small sample size of 50 patients, we believe it provides valuable insights into the potential long-term cardiac effects of COVID-19. Specifically, this study is among the beginning observations focused on GLS impairment as a marker of myocardial dysfunction in asymptomatic patients. We acknowledge that larger, prospective studies are needed to confirm these findings, but we consider this work an important preliminary step in highlighting this issue.
We added in the introduction section, specifying that: “These results could contribute to the characterization of post-COVID syndrome in the South-Eastern region of Romania, as no previous reports on this subject are available.”
We added as a limit of the study: “The sample size of the pilot-study is relatively small and statistical significance of the results could be in short supply.”
- Comorbidities and Follow-Up: We understand your concerns regarding the impact of comorbidities over the two years of follow-up, which led to an 18% reduction in LVEF.
Before initiating this study, we reviewed the existing literature on COVID-19 and its myocardial involvement. Our findings revealed that most studies primarily focused on the initial months following diagnosis, without differentiating between the severity of the disease. Based on this, we hypothesized the following:
- Patients with severe COVID-19 have a higher likelihood of cardiac involvement.
- This cardiac involvement could be long-lasting—i.e., it may persist beyond the recovery phase.
Consequently, we began an evaluation of these patients using echocardiography. This assessment was conducted more than 24 months after their initial hospitalization, which, we believe, is the key strength of our study, as most other studies have assessed patients much earlier in the post-acute phase.
Regarding traditional cardiovascular risk factors, which might confound our findings, it is important to note that most of our patients were young (<65 years, 54%), less than half had hypertension, and most did not have diabetes, heart disease, or chronic kidney disease.
3. Contribution to the Literature: Despite its limitations, this study adds to the understanding of the potential long-term cardiovascular impact of COVID-19.
Let me know if you'd like me to tailor this further!
Kind regards,
Manuela Arbune
Reviewer 2 Report
Comments and Suggestions for Authors
The COVID-19 pandemic has transformed health systems worldwide. There is conflicting data regarding the degree of cardiovascular involvement following infection. The studies were designed to evaluate the prevalence of echocardiographic abnormalities in adults recovered from COVID-19. In Garcia-Zamora S. et al. study was shown that the most frequent findings were a decrease in the global longitudinal strain (GLS) of both the left and the right ventricle, followed by a mild reduction in the left ventricular ejection fraction [Garcia-Zamora S, Picco JM, Lepori AJ, Galello MI, Saad AK, Ayón M, Monga-Aguilar N, Shehadeh I, Manganiello CF, Izaguirre C, Fallabrino LN, Clavero M, Mansur F, Ghibaudo S, Sevilla D, Cado CA, Priotti M, Liblik K, Gastaldello N, Merlo PM. Abnormal echocardiographic findings after COVID-19 infection: a multicenter registry. Int J Cardiovasc Imaging. 2023;39(1):77-85. doi: 10.1007/s10554-022-02706-9.].
The aim of this study was to determine the frequency of echocardiographic changes observed two years after hospitalization for moderate and severe COVID-19.
Based on the study results, authors revealed main findings. There are: 1) three year post-COVID-19 patients with no prior cardiac disease revealed a high prevalence of LV dysfunction, primarily diagnosed based on a decrease in GLS, suggesting myocardial contractile dysfunction even in patients with asymptomatic heart disease; 2) systemic inflammation and hemodynamic changes associated with pulmonary impairment during COVID-19 may affect myocardial function, although no correlations were found between infection severity markers and left ventricle dysfunction; 3) GLS could serve as a screening marker for better understanding the spectrum of post-COVID-19 cardiological impairment; 4) the mechanisms of post-COVID-19 myocardial impairment are complex, overlap with natural dysfunction, remain insufficiently clarified, and require long-term, extensive epidemiological studies; 5) regular echocardiographic monitoring of patients recovering from moderate to severe COVID-19 should be systematically conducted to enable early detection of myocardial dysfunction and assess the risk of progression to heart failure.
The findings are consistent with the evidence and arguments presented. All the main questions raised were considered. In general manuscript is written properly. But at the same time, study has some limitations. The main limitations are absence of baseline (prior) echocardiographic evaluations before the acute COVID-19 episode, lack of a control group comprising healthy individuals and miss of echocardiographic indicators comparison with MRI parameters. Other limitations are single-center, small sample size, nonrandomized and retrospective study.
References are appropriate. 42 (100.0%) references are given, of which 40 (95.2%) are less than 5 years old. Article is not including an excessive number of self-citations.
The manuscript is scientifically sound and the experimental design is appropriate to test the hypothesis.
The manuscript’s results are reproducible based on the details given in the methods section.
The 2 figure, 5 tables and 3 additional tables are presented. The figure and tables are appropriate. They are properly showed the data. They are easy to interpret and understand. The data is interpreted appropriately and consistently throughout the manuscript.
The conclusions are consistent with the evidence and arguments presented.
The ethics statement is appropriate.
The data availability is appropriate.
Specific comments:
1. The authors should remove the phrases “global longitudinal strain (GLS) of the left ventricle (LV)” in Discussion second paragraph. Because they have already been cited above. I advise authors to use only abbreviations: GLS and LV, respectively.
Author Response
Dear Reviewer,
Thank you very much for your time in carefully reviewing the manuscript and for appreciating the results of our work on this study.
We are grateful for your kindness and favorable comments.
Comments and Suggestions for Authors
The COVID-19 pandemic has transformed health systems worldwide. There is conflicting data regarding the degree of cardiovascular involvement following infection. The studies were designed to evaluate the prevalence of echocardiographic abnormalities in adults recovered from COVID-19. In Garcia-Zamora S. et al. study was shown that the most frequent findings were a decrease in the global longitudinal strain (GLS) of both the left and the right ventricle, followed by a mild reduction in the left ventricular ejection fraction [Garcia-Zamora S, Picco JM, Lepori AJ, Galello MI, Saad AK, Ayón M, Monga-Aguilar N, Shehadeh I, Manganiello CF, Izaguirre C, Fallabrino LN, Clavero M, Mansur F, Ghibaudo S, Sevilla D, Cado CA, Priotti M, Liblik K, Gastaldello N, Merlo PM. Abnormal echocardiographic findings after COVID-19 infection: a multicenter registry. Int J Cardiovasc Imaging. 2023;39(1):77-85. doi: 10.1007/s10554-022-02706-9.].
The aim of this study was to determine the frequency of echocardiographic changes observed two years after hospitalization for moderate and severe COVID-19.
Based on the study results, authors revealed main findings. There are: 1) three year post-COVID-19 patients with no prior cardiac disease revealed a high prevalence of LV dysfunction, primarily diagnosed based on a decrease in GLS, suggesting myocardial contractile dysfunction even in patients with asymptomatic heart disease; 2) systemic inflammation and hemodynamic changes associated with pulmonary impairment during COVID-19 may affect myocardial function, although no correlations were found between infection severity markers and left ventricle dysfunction; 3) GLS could serve as a screening marker for better understanding the spectrum of post-COVID-19 cardiological impairment; 4) the mechanisms of post-COVID-19 myocardial impairment are complex, overlap with natural dysfunction, remain insufficiently clarified, and require long-term, extensive epidemiological studies; 5) regular echocardiographic monitoring of patients recovering from moderate to severe COVID-19 should be systematically conducted to enable early detection of myocardial dysfunction and assess the risk of progression to heart failure.
The findings are consistent with the evidence and arguments presented. All the main questions raised were considered. In general manuscript is written properly. But at the same time, study has some limitations. The main limitations are absence of baseline (prior) echocardiographic evaluations before the acute COVID-19 episode, lack of a control group comprising healthy individuals and miss of echocardiographic indicators comparison with MRI parameters. Other limitations are single-center, small sample size, nonrandomized and retrospective study.
References are appropriate. 42 (100.0%) references are given, of which 40 (95.2%) are less than 5 years old. Article is not including an excessive number of self-citations.
The manuscript is scientifically sound and the experimental design is appropriate to test the hypothesis.
The manuscript’s results are reproducible based on the details given in the methods section.
The 2 figure, 5 tables and 3 additional tables are presented. The figure and tables are appropriate. They are properly showed the data. They are easy to interpret and understand. The data is interpreted appropriately and consistently throughout the manuscript.
The conclusions are consistent with the evidence and arguments presented.
The ethics statement is appropriate.
The data availability is appropriate.
Specific comments:
- The authors should remove the phrases “global longitudinal strain (GLS) of the left ventricle (LV)” in Discussion second paragraph. Because they have already been cited above. I advise authors to use only abbreviations: GLS and LV, respectively.
Response:
We removed.
Kind regards,
Manuela Arbune
Submission Date
16 December 2024
Date of this review
24 Dec 2024 06:31:0
Round 2
Reviewer 1 Report
Comments and Suggestions for Authors
I have no many comments to do since the paper was already and previously rejected. Limitations of paper are clear
Author Response
Dear Reviewer,
Thank you very much for your time in carefully reviewing the manuscript and for your thoughtful feedback raising important points. We appreciate the opportunity to clarify and address these concerns.
Comments and Suggestions for Authors
Authors conclude that COVID-19 is an independent predictive factor for GLS impairment, which can indicate myocardial contractile dysfunction, even in patients with asymptomatic heart disease. However, I am not sure this retrospective study over only 50 patients may add something to medical literature, also considering comorbidities over 2 years of follow-up, leading to 18% of reduction of Left ventricular ejection fraction (LVEF).
Respond:
Sample size and study design
While our pilot study includes a relatively small sample size of 50 patients, we believe it provides valuable insights into the potential long term cardiac effects od COVID-19. Specifically, this study is among the beginning observations focused on GLS impairment as a marker of myocardial dysfunction in asymptomatic patients. We acknowledge that larger, prospective studies are needed to confirm these findings, but we consider this work an important preliminary step in highlighting this issue.
We added in the introduction section, specifying that: “These results could contribute to the characterization of post-COVID syndrome in the South-Eastern region of Romania, as no previous reports on this subject are available.”
We added as a limit of the study: “The sample size of the pilot-study is relatively small and statistical significance of the results could be in short supply.”
Comorbidities and follow-up
We understand your concerns regarding the impact of comorbidities over the two years of follow-up, which led to an 18% reduction in LVEF. Before initiating this study, we reviewed the existing literature on COVID-19 and its myocardial involvement. Our findings revealed that most studies primarily focused on the initial months following diagnosis, without differentiating between the severity of the disease. Based on this, we hypothesized that patients with severe COVID-19 have higher likelihood of cardiac involvement that could be long lasting and may persist beyond the recovery phase.
Consequently, we began an evaluation of these patients using echocardiography. This assessment was conducted more than 24 months after the initial hospitalization, witch, we believe, is the key strength of our study, as most other studies have assessed patients much earlier in the post-acute phase.
Regarding traditional cardiovascular risk factors, witch might confound our findings, it is important to note that most our patients were young (<65 years, 54%), less than a half had hypertension, and most did not have diabetes, heart disease, or chronic kidney disease.
Contribution to the literature
Despite its limitations, this study adds to the understanding of the potential long-term cardiovascular impact of COVID-19. Considering some significant findings, we believe our study serves as a “Proof of concept” that subclinical myocardial involvement can persist long after the initial diagnosis, regardless of the disease’s initial severity.
First, there was no correlation between the severity of COVID-19 and subclinical myocardial disease. These findings may seem counterintuitive, as severe forms of COVID-19 are typically associated with a higher risk of cardiovascular disease.
Secondly, when evaluating patients based on impaired GLS and/or impaired filing, we found that 40,8% met the criteria for subclinical left ventricular impairment. Notably, this evaluation was conducted more than two years after the initial COVID-19 diagnosis in patients without significant cardiovascular risk factors.
However, whether this finding correlates with poor clinical outcomes remains to be determined in future studies.
Future directions
We agree that larger, prospective studies are warranted to further elucidate the long-term cardiac effects of COVID-19, particularly in relation to comorbidities and other confounding factors. We hope our study serves as a starting point for further investigations.
Thank you again!
Kind regards,
Manuela Arbune
